# Longitudinal stability and interrelations between health behavior and subjective well-being in a follow-up of nine years

**Säde Stenlund**[1,2]*, **Niina Junttila**[3], **Heli Koivumaa-Honkanen**[4,5], **Lauri Sillanmäki**[1,2,6], **David Stenlund**[7], **Sakari Suominen**[1,2,8], **Hanna Lagström**[1,2,9], **Päivi Rautava**[1,2]

**1** Department of Public Health, University of Turku, Turku, Finland, **2** Research Services, Turku University Hospital, Turku, Finland, **3** Department of Teacher Education, University of Turku, Turku, Finland, **4** Institute of Clinical Medicine, Psychiatry, University of Eastern Finland, Kuopio, Finland, **5** Mental Health and Wellbeing Center, Kuopio University Hospital, Kuopio, Finland, **6** Department of Public Health, University of Helsinki, Helsinki, Finland, **7** Faculty of Science and Engineering, Åbo Akademi University, Turku, Finland, **8** School of Health Sciences, University of Skövde, Skövde, Sweden, **9** Centre for Population Health Research, University of Turku and Turku University Hospital, Turku, Finland

* sade.stenlund@utu.fi

**Data Availability Statement:** We kindly notify that study data contains personal and sensitive

## Abstract

### Background

The bidirectional relationship between health behavior and subjective well-being has previously been studied sparsely, and mainly for individual health behaviors and regression models. In the present study, we deepen this knowledge focusing on the four principal health behaviors and using structural equation modeling with selected covariates.

### Methods

The follow-up data (n = 11,804) was derived from a population-based random sample of working-age Finns from two waves (2003 and 2012) of the Health and Social Support (HeSSup) postal survey. Structural equation modeling was used to study the cross-sectional, cross-lagged, and longitudinal relationships between the four principal health behaviors and subjective well-being at baseline and after the nine-year follow-up adjusted for age, gender, education, and self-reported diseases. The included health behaviors were physical activity, dietary habits, alcohol consumption, and smoking status. Subjective well-being was measured through four items comprising happiness, interest, and ease in life, and perceived loneliness.

### Results

Bidirectionally, only health behavior in 2003 predicted subjective well-being in 2012, whereas subjective well-being in 2003 did not predict health behavior in 2012. In addition, the cross-sectional interactions in 2003 and in 2012 between health behavior and subjective well-being were statistically significant. The baseline levels predicted their respective follow-up levels, the effect being stronger in health behavior than in subjective well-being.

information and due to the present legislation in Finland cannot be made publicly available inside or outside Finland. Data inquiries may be sent to Prof. Markku Sumanen (markku.sumanen@tuni.fi), leader of the HeSSup research group. Additionally, Lauri Sillanmäki (lauri.sillanmaki@utu.fi) is responsible for the data storage and can be contacted concerning data inquiries. Due to the national data protection legislation, the data used in this study cannot be shared without applying for permission to use the data with a specific study protocol and scientifically justified study questions.

**Funding:** This work was supported by personal grants by the Signe and Ane Gyllenberg Foundation (SSt:5175) and (HKH:5525) [2020], https:// gyllenbergs.fi/fi/apurahat-ja-symposiumit; the Waldemar von Frenckell Foundation (SSt) [2020], http://www.foundationweb.net/frenckell/; the Päivikki and Sakari Sohlberg Foundation (HKH:2020), https://pss-saatio.fi; and the Magnus Ehrnrooth Foundation (DS) [2020], https://www. magnusehrnroothinsaatio.fi/en/. The funding sources did not participate in designing or conducting the study; collection, management, analysis or interpretation of the data; or preparation, review, or approval of the manuscript.

**Competing interests:** The authors have declared that no competing interests exist.

## Conclusion

The four principal health behaviors together predict subsequent subjective well-being after an extensive follow-up. Although not particularly strong, the results could still be used for motivation for health behavior change, because of the beneficial effects of health behavior on subjective well-being.

## Introduction

The cross-sectional association between health behavior and various measures of subjective well-being is well-established [1–4], but the longitudinal associations need further research [5]. Only a few studies have focused on them [6]. In a follow-up of 15 years, a bidirectional relationship was evident between adverse alcohol consumption and life dissatisfaction, but the former was a somewhat stronger predictor of the latter than vice versa in a large sample of adults [7]. Fruit and vegetable consumption predicted better life satisfaction in a two-year follow-up when adjusted for other health behaviors, but not vice versa [8]. Positive changes in dietary patterns [8,9] and physical activity [10] have resulted in better life satisfaction. In addition, measures of subjective well-being have predicted greater reduction in smoking [11] and less relapses [12]. Earlier reviews have stated that the effect of subjective well-being on subsequent health behavior is largely explained by baseline health behavior [13] but that the bidirectional nature of the relationship requires further study [14]. Subjective well-being refers to a personal evaluation and appraisal of one's life including both a cognitive judgement (such as life satisfaction) and an emotional response on life (such as happiness) [15].

Structural equation modeling is suitable for health behavior research due to the possibility of including multiple causes and outcomes, lower risk of type I error compared to univariate or bivariate testing, the possibility to specify relationships between variables, reduced effect of measurement error, and advanced treatment of missing data [16]. It enables a more detailed analysis of individual components of health behavior and subjective well-being as their distinctive impacts on the latent variable are investigated. In a study using structural equation modeling [17], the effects of smoking on subsequent lower life satisfaction, lower optimism, and less positive affect (path coefficients = 0.10–0.025) were stronger than the effects in the opposite direction (path coefficients = 0.04–0.08) in a four-year follow-up on older adults (mean age = 64 years). The cross-sectional associations between smoking and subjective well-being at baseline were also statistically significant (path coefficients = 0.04–0.05). Nevertheless, the strongest path coefficients were observed between baseline and follow-up for either smoking (1.77) or subjective well-being (0.43–0.64). To the best of our knowledge, however, the associations between subjective well-being and multiple health behaviors or single health-promoting behaviors, such as physical activity and dietary habits, have not previously been studied using structural equation modeling.

The aim of the present study was to explore the cross-sectional, longitudinal, and cross-lagged relationships of health behavior and subjective well-being by structural equation modelling, as shown in Fig 1. Our hypothesis was that health behavior predicts subsequent subjective well-being and vice versa, because bidirectionality of the relationship has been suggested earlier [14]. Based on earlier research, we also anticipated that the cross-sectional relationships between health behavior and subjective well-being would be statistically significant. Furthermore, we presumed that health behavior at baseline would be a significant predictor of health

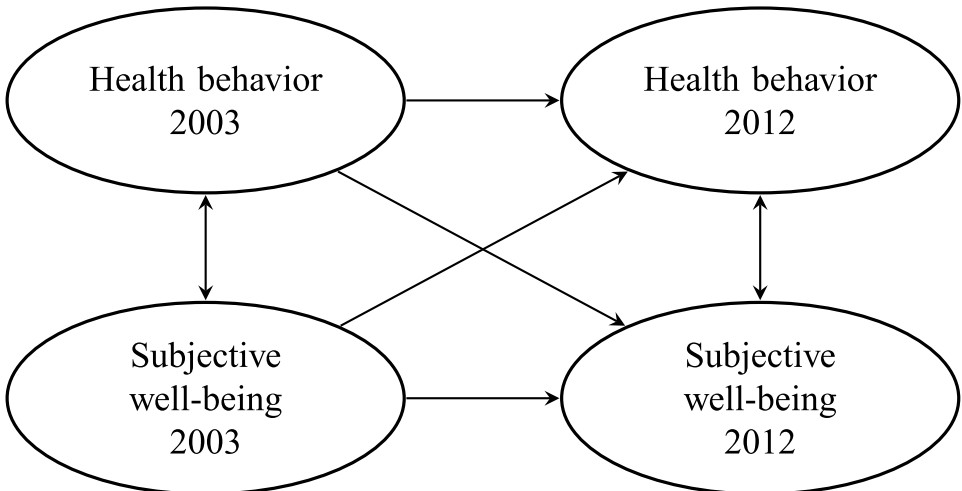

**Fig 1. Structural model showing the connections between latent variables that were tested in the analysis.**

behavior at follow-up, as well as subjective well-being predicting subsequent subjective well-being.

## Methods

The data used in this study comes from a random sample (n = 13,050) of the Finnish working-age population that was collected in the Health and Social Support (HeSSup) study. We used postal survey data from the second (2003) and the third wave (2012) of the study. The survey questions were identically phrased and were presented in an identical or almost identical order. Individuals with missing information on any of the covariates (n = 1,244) were excluded from the analysis. The selection process of the study population is outlined in Fig 2. The final study population (n = 11,806) did not differ considerably from the respondents of the two waves (n = 13,050) with regard to age and gender. For details of the comparison, see S1 File.

Four age groups were represented: 25–29 years; 35–39 years; 45–49 years; 55–59. When the HeSSup-study commenced in 1998, the age groups were chosen to be non-continuous to capture certain life transition periods by creating clearly distinct groups that would be fairly homogeneous, i.e., ages not spreading over an entire decade. Four educational groups were represented: no professional education; vocational course/school/apprenticeship contract; college; university degree/university of applied sciences. Major diseases from a pre-defined list of 32 conditions were grouped as follows: none; one; two or more.

The concurrent joint Ethical Committee of the University of Turku and the Turku University Central Hospital approved the Health and Social Support study. The present study was carried out according to the Declaration of Helsinki. Participants signed a written consent agreeing to a prospective follow-up including the registry data.

### Measures

**Health behavior.**   Data on four principal health behaviors were dichotomized with the two categories beneficial and risky behavior. *Physical activity* was measured by metabolic equivalent task (MET) where a score of 2, corresponding to 30 minutes of walking per day, was the cut-off point between active and inactive [18]. *Dietary habits* were dichotomized at the median of a dietary index (range 0–10) measuring adherence to Nordic nutritional

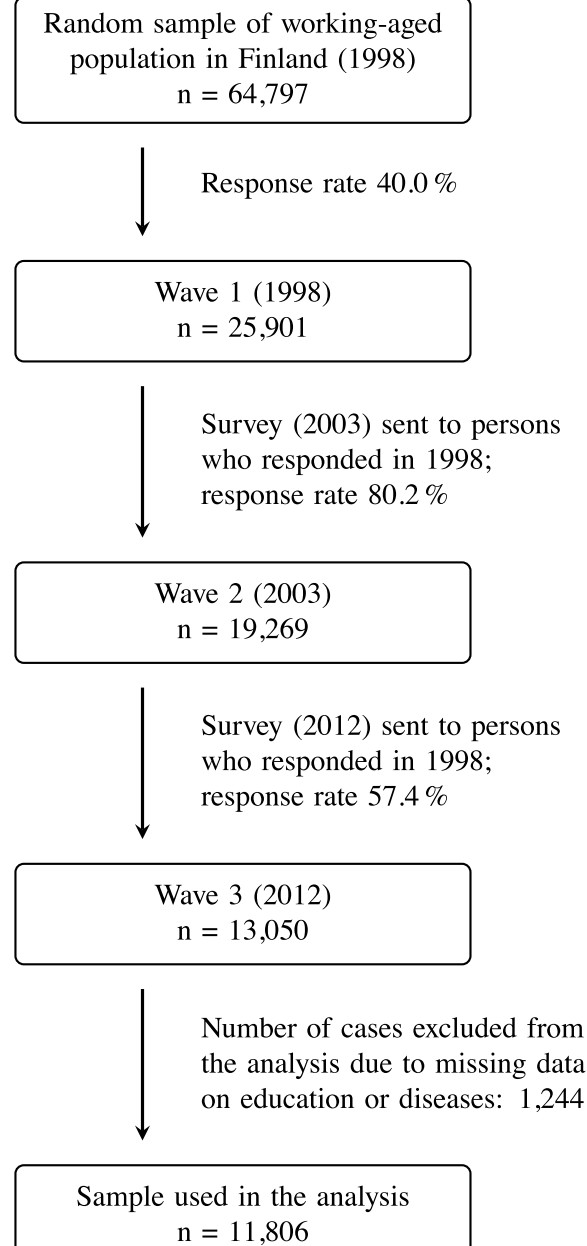

**Fig 2. Outline of population data selection.** Inclusion and exclusion criteria for the present study population from the Health and Social Support (HeSSup) prospective population-based follow-up study.

recommendations [19] for ten food items in a non-validated food questionnaire. Each choice that was in line with the following recommendations contributed to one point: dark bread ($\geq$ 2/day); pastries and sweets ($\leq$ 1–2/week); fat free milk ($\geq$ 1/day); sausages ($\leq$ 1–2/week); red meat ($\leq$ 1–2/week); chicken or turkey ($\leq$1–2/week); fish ($\geq$ 1–2/week); fresh fruits and berries ($\geq$ 2/day); vegetables ($\geq$ 2/day); alcohol use ($<$ 10g/day women, 20g/day men) [20]. *Alcohol consumption* was dichotomized according to the at-risk use level in Finland, excessive consumption being $\geq$ 140g/week for women and $\geq$ 280g/week for men [21]. *Smoking status* divided participants into current smokers and others.

**Subjective well-being.** Items from a four-item life satisfaction scale [22] were used to reflect subjective well-being with a reversed scale (i.e. higher scores indicating better life satisfaction). The items assessed interest (1–5), happiness (1–5), and ease in life (1–5), and perceived loneliness (1–4) as follows: very boring/unhappy/hard/lonely = 1; fairly boring/unhappy/hard/lonely = 2; cannot say = 3; fairly interesting/happy/easy and not at all lonely = 4; very interesting/happy/easy = 5 [7,22]. On the original scale, perceived loneliness ranged from 1 to 5 but had no response alternative 2. The scale was compressed into a scale ranging from 1 to 4.

In general, subjective well-being comprises both cognitive and affective components [23]. Thus, the four items in the life satisfaction scale can be regarded to reflect subjective well-being–as we have assumed here–rather than solely that of life satisfaction, which has been defined as the cognitive component of subjective well-being [5]. It is also more appropriate to let its four components represent subjective well-being in the structural equation model, in which the components represent distinct observed variables and, thus, different aspects of subjective well-being.

## Statistical analysis

The hypothesized models were tested using structural equation modeling with the MPlus software, version 7.4 [24]. The analyses, based on the covariance matrices of the data, were performed using the mean- and variance-adjusted weighted least squares estimator (WLSMV). This estimator was used because the observed health behavior variables were categorical. The fit of the models was evaluated using the following indexes (levels of acceptable fit): the comparative fit index (CFI > 0.90), the Tucker-Lewis Index (TLI > 0.90), and the root mean square error of approximation (RMSEA < 0.08) [25]. The $\chi^2$ values were not primarily used for model fit estimation, because a good fit using $\chi^2$ can be hard to achieve in large samples [26], but the $\chi^2$ values were still used for comparison between different models.

The latent variables used in our model were health behavior and subjective well-being, each at two time points (2003 and 2012). The structural validity of these latent variables in 2003 was confirmed through confirmatory factor analysis (CFA), which tests the adequacy of the specified relations between the constructed latent variables and their corresponding indicators [27]. Single items and errors were assumed to be uncorrelated. The fit of the CFA model was good, with CFI = 0.990, TLI = 0.985 and RMSEA = 0.019 in 2003, and only slight differences in these values were observed in 2012. When performing CFA separately on the four individual latent variables, the fit was acceptable in all cases, except that TLI was somewhat under the cut-off value (TLI = 0.887) for health behavior in 2003. Despite this slight discordance, no further modifications on the health behavior latent variable were made, as the CFA with both latent variables included showed good fit in 2003 and also in 2012, and the goal of the present study was to explore the relationship of the four principal health behaviors with subjective well-being.

To study the bidirectional relationships between health behavior and subjective well-being in 2003 and 2012, a latent variable structural model was constructed as shown in Fig 1. For the model to be estimated, autocorrelations of the individual components of health behavior and subjective well-being were included. This crude model was tested for the respondents of the two waves (n = 13,050) and for the population used in the final model (n = 11,806), which confirmed that the omission of participants having missing data on covariates did not substantially alter the results. For details, see S1 File. Thereafter, the covariates gender, age, education, and diseases were included to serve as potential predictors of health behavior and subjective well-being. The model adjusted for the covariates showed good fit: CFI = 0.958, TLI = 0.944, and

RMSEA = 0.035. The MPlus software suggested the following connections as potential modifications to the model: physical activity with alcohol consumption, dietary habits with smoking, and interest in life with ease of living. These connections were excluded after testing, however, since the fit of the model did not improve substantially.

A discussion on reliability measures is included in S2 File. Since the assumption of tau-equivalence does not hold, McDonald's ω is preferable over the commonly used Cronbach's α as a measure of reliability. Values of both α and ω are given in Table A in S2 File.

## Results

We explored the bidirectional longitudinal relationships between health behavior and subjective well-being, when concomitant influence of several confounders was taken into consideration, due to the applied statistical method of analysis. The baseline characteristics of the participants are presented in Table 1. The final model is shown in Fig 3, where all indicated connections are statistically significant (p < 0.001). Health behavior in 2003 predicted subjective well-being in 2012 by a standardized path coefficient of 0.156, but subjective well-being in 2003 did not show a statistically significant prediction for health behavior in 2012. However, both predicted (in 2003) their own subsequent levels (in 2012), with the standardized path coefficients being 0.896 for health behavior and 0.468 for subjective well-being. Their cross-sectional associations were also significant, the standardized path coefficients being 0.302 in 2003 and 0.159 in 2012. The path coefficients and R-values are presented in Table 2, the descriptive means of the observed variables in Table 3, and the changes in fit indexes adjusted for covariates and suggested modifications in Table 4. Table 5 summarizes the path coefficients and autocorrelations of the observed variables.

## Discussion

Our study on 11,800 working-age Finns was focused on the association between four principal health behaviors and subjective well-being as latent variables in a structural equation model adjusted for age, gender, education, and major diseases at baseline. The results suggest that health behavior predicts subjective well-being after a nine-year follow-up at a weak but still significant level, but not vice versa. Baseline health behavior is a strong determinant of

**Table 1. Baseline characteristics of the participants in 2003.**

| Variable | Category | Share of the study population % (n) |
|---|---|---|
| Study population | | 100 (11,806) |
| Age | 25–29 | 20.7 (2,449) |
| | 35–39 | 20.5 (2,422) |
| | 45–49 | 26.7 (3,155) |
| | 55–59 | 32.0 (3,780) |
| Gender | Male | 37.1 (4,382) |
| | Female | 62.9 (7,424) |
| Education | No professional education | 12.2 (1,436) |
| | Vocational school | 28.7 (3,391) |
| | College | 39.0 (4,599) |
| | University level education | 20.2 (2,380) |
| Diseases | 0 | 18.0 (2,129) |
| | 1 | 23.4 (2,759) |
| | 2 or more | 58.6 (6,918) |

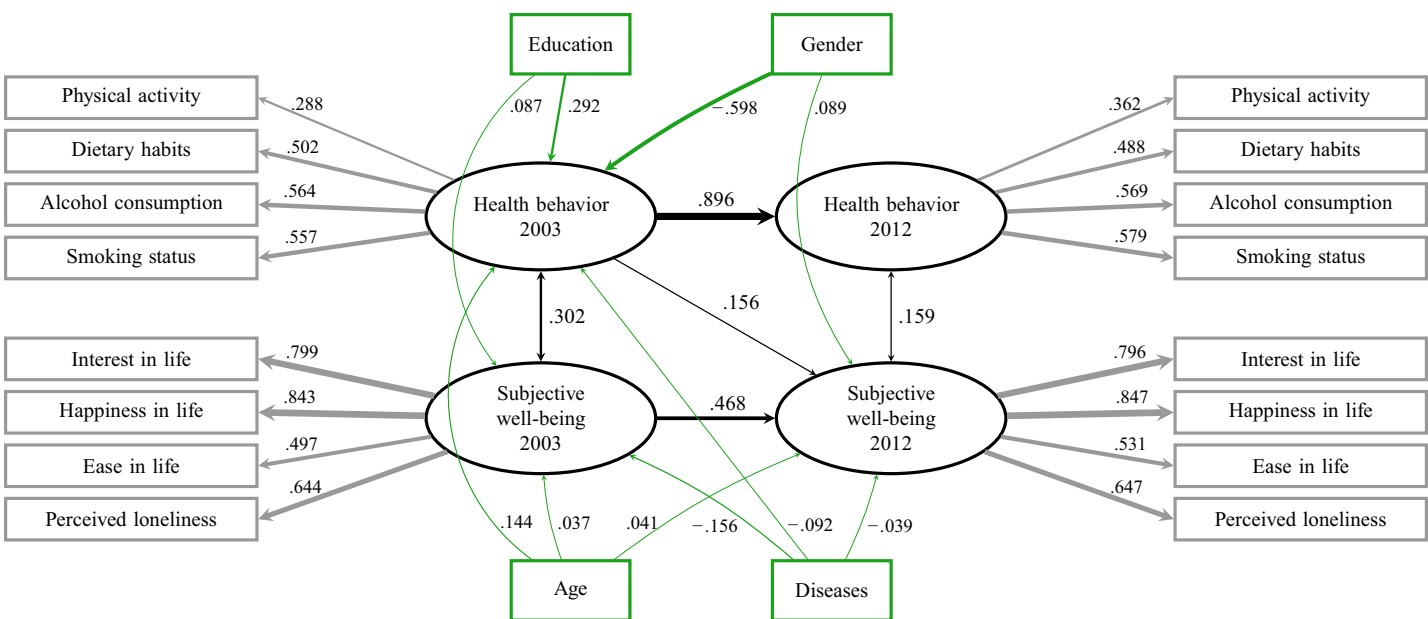

**Fig 3. Path diagram of final structural equation model.** The structural equation model showing standardized path coefficients for cross-lagged, cross-sectional, and longitudinal connections for health behavior and subjective well-being in a nine-year follow-up adjusted for covariates.

subsequent health behavior and baseline subjective well-being is a moderate determinant of subsequent subjective well-being. The cross-sectional associations of health behavior and subjective well-being are evident.

Our study contributes to the understanding of the relationship between health behavior and subjective well-being. Health behavior is highly correlated with its subsequent level, which indicates that it is a stable characteristic. In addition, 79% of its subsequent level can be accounted for by the factors in our model. Physical activity seems to be the least strongly

**Table 2. Standardized path coefficients and R-values in the structural equation model.**

|  | HB$_{2003}$ | SWB$_{2003}$ | HB$_{2012}$ | SWB$_{2012}$ | R |
|---|---|---|---|---|---|
| **HB$_{2003}$** |  | 0.302 | 0.896 | 0.156 | 0.183 |
| **SWB$_{2003}$** |  |  | ns | 0.468 | 0.032 |
| **HB$_{2012}$** |  |  |  | 0.159 | 0.791 |
| **SWB$_{2012}$** |  |  |  |  | 0.294 |
| **Age** | 0.144 | 0.037 | ns | 0.041 |  |
| **Gender**[a] | −0.598 | ns | ns | 0.089 |  |
| **Education** | 0.292 | 0.087 | ns | ns |  |
| **Diseases** | −0.092 | −0.156 | ns | −0.039 |  |

Estimates for health behavior, subjective well-being and covariates based on data from the Health and Social Support (HeSSup) study.

HB$_{2003}$ = Health behavior in 2003.

HB$_{2012}$ = Health behavior in 2012.

SWB$_{2003}$ = Subjective well-being in 2003.

SWB$_{2012}$ = Subjective well-being in 2012.

ns = non-significant, for all other values p < 0.001.

CFI = 0.958, TLI = 0.944, and RMSEA = 0.035.

[a] Gender is a binary covariate (female = 0, male = 1), and the values are therefore STDY standardization, while all other values are STDYX.

**Table 3. Means (standard deviations) of observed variables in the Health and Social Support (HeSSup) study.**

| Latent variable | Observed variable | 2003 | 2012 |
|---|---|---|---|
| Health behavior Risky = 1 Beneficial = 2 | Physical activity (1 or 2) | 1.72 | 1.71 |
| | Dietary habits (1 or 2) | 1.39 | 1.47 |
| | Alcohol consumption (1 or 2) | 1.95 | 1.95 |
| | Smoking status (1 or 2) | 1.81 | 1.86 |
| Subjective well-being | Interest in life (1–5) | 3.90 (0.93) | 3.88 (0.92) |
| | Happiness in life (1–5) | 3.97 (0.83) | 3.98 (0.82) |
| | Ease of living (1–5) | 3.47 (1.02) | 3.63 (0.99) |
| | Not feeling lonely (1–4) | 3.43 (0.91) | 3.46 (0.89) |

reflected behavioral mode among the observed indicators of health behavior. Alcohol consumption and smoking seem to be the most stable characteristics. For subjective well-being, the factor loadings are higher than for health behavior. This suggests that the components of subjective well-being are more consistently co-varying than those of health behavior, which is also reflected in Cronbach's alphas being higher for subjective well-being than for health behavior. Note, however, the caveats of using Cronbach's alpha discussed in S2 File. However, subjective well-being was a less stable characteristic during follow-up than the latent variable of health behavior when using continuous measures for subjective well-being and dichotomized for health behavior. Happiness and interest in life have the strongest impact on the latent variable. About 30% of the subsequent level of subjective well-being was determined by factors in our model, even when the model included baseline subjective well-being. Thus, subjective well-being is not primarily determined by factors in our model, but largely by factors outside it. The results also suggest that ease of living has less impact on subjective well-being than the other measures. Significant cross-sectional correlations between health behavior and subjective well-being are observed, which could partly be caused by external factors and partly by their effect on each other.

Being older, being a woman, having higher education, and having less diseases were associated with somewhat better health behavior at baseline. However, the effect was not present at the follow-up. This could be explained by the fact that the covariates account for interindividual differences and do not substantially change during follow-up. Therefore, the effect of covariates is included in the effect that baseline health behavior has on subsequent health behavior. Diseases at baseline are associated with both worse health behavior and subjective well-being at baseline and worse subjective well-being at the follow-up. Men showed slightly better subjective well-being at the follow-up. Better education has positive impact both on the baseline subjective well-being and on health behavior. Age has an impact on subjective well-being at both time points and on health behavior at baseline, which is in line with earlier research [15].

**Table 4. Model fit statistics on the bidirectional longitudinal association between health behavior and subjective well-being.**

| Model | $\chi^2$ (df) | CFI/TLI | RMSEA |
|---|---|---|---|
| Model 1: Crude model | 432 (90) | .992/.990 | .017 |
| Model 2: Age, gender, education, major diseases included[a] | 2079 (138) | .958/.944 | .035 |
| Model 3: Suggested modifications[b] added to Model 2 | 1966 (132) | .960/.944 | .034 |

[a] Final model, see Fig 1.

[b] Suggested modifications were connections between physical activity and alcohol consumption, dietary habits and smoking, as well as interest in life and ease of living.

**Table 5. Standardized path coefficients, correlations, and autocorrelations of observed variables.**

| Observed variable | 2003 | 2012 | Autocorrelation | Correlation 2003–2012 |
|---|---|---|---|---|
| Physical activity | 0.288 | 0.362 | 0.415 | 0.492 |
| Dietary habits | 0.502 | 0.488 | 0.291 | 0.477 |
| Alcohol consumption | 0.564 | 0.569 | 0.510 | 0.757 |
| Smoking status | 0.557 | 0.579 | 0.669 | 0.917 |
| Interest in life | 0.799 | 0.796 | 0.076 | 0.412 |
| Happiness in life | 0.843 | 0.847 | 0.001 (ns) | 0.363 |
| Ease of living | 0.497 | 0.531 | 0.169 | 0.306 |
| Not feeling lonely | 0.644 | 0.647 | 0.153 | 0.399 |

ns = non-significant.

All path coefficients and autocorrelations are statistically significant (p < 0.001) unless indicated otherwise.

In another study using structural equation modelling, non-smoking predicted measures of subjective well-being and vice versa in older adults (mean age = 64 years) with a follow-up of four years [17]. The highest path coefficient was observed for smoking status predicting subsequent smoking (1.77), which is in line with our study showing high longitudinal correlations of health behaviors and smoking having the highest value of these. Similar to our model, the magnitudes of the path coefficients from baseline subjective well-being to subsequent subjective well-being at follow-up were higher (0.43–0.64) than for the cross-sectional (0.04–0.05 or non-significant) or cross-lagged associations (0.04–0.25 or non-significant). The magnitudes of the path coefficients from smoking to subsequent subjective well-being were higher (0.10–0.25) than vice versa (0.04–0.08 or non-significant), which also is in line with the results in our study. However, our study observed stronger cross-sectional associations than cross-lagged associations. There may be multiple reasons for this disparity. The follow-up was shorter in the study by Lappan, which might result in stronger longitudinal associations. Inclusion of multiple health behaviors could also strengthen the concurrent bidirectional effect to subjective well-being; exercising and eating a healthy diet could promote subjective well-being, and higher levels of subjective well-being could promote self-efficacy to maintain healthy behavior. The age difference in the studies can also have an effect; health behavior of working-aged persons is perhaps guided more by lack of time compared to that of older adults, for whom other factors such as emerging health conditions might be more dominant.

## Implications

The results demonstrate the stability of health behavior, which underlines the importance of targeting health behavior early in life when health behavior patterns are formed. Health behavior shows a longitudinal association with subjective well-being, which could serve as a motivator for health behavior change on an individual level. The results could also emphasize the relevance of political actions targeting health behavior, not only to reduce the increasing costs of non-communicable diseases, but also to maintain and improve people's subjective well-being. Good mental health and subjective well-being of citizens are valuable goals in itself, but they can also lead to greater productivity [28]. The results could be generalized to the whole working-age population in Finland and presumably to corresponding populations of other Western countries.

## Evaluation

The use of structural equation modeling enables deeper and more detailed understanding of the bidirectional association between health behavior and subjective well-being than previous

research has provided. The large population-based sample yields solid results. Many of the challenges in structural equation modeling [27] were considered by the following strategies: large population-based random sampling, longitudinal study set-up, use of multiple fit indexes, inclusion of autocorrelations, theory-based and limited use of modifications and covariates. However, only a limited number of models were tested. This was partly done because of the interest to deepen the understanding from earlier research and partly because the observed variables were based on earlier literature. Including autocorrelation accounted for measurement error, which is of importance especially for the continuous measure of subjective well-being, resulting in more reliable estimates of variance.

The association between health behavior and subsequent subjective well-being was statistically significant but fairly weak. However, its strength was about a third of the path coefficient of how subjective well-being predicted its subsequent level and of similar magnitude to the results in the study by Lappan et al. [17] exploring the relationship between smoking and measures of subjective well-being. The four principal health behaviors associated to the risk of chronic diseases of public health concern [29,30] were the focus of this study. However, their factor loadings varied considerably. Different components of health behavior or individual health behaviors could yield more reliable factors and results. The measures for subjective well-being were items from the four-item life satisfaction scale [22], which as an indicator of subjective well-being reflects its cognitive and affective components. It could be argued that loneliness is not part of the standard subjective well-being, but it was included because of a good fit in the factor analysis and a significant path coefficient. Health behavior was a more stable characteristic than subjective well-being, but it was measured by dichotomized variables, where small changes remain unrecorded compared to the scales for subjective well-being. The study was conducted in Finland, where the social security system is strong and for example gender or economic inequality low. Therefore, the results are most reliably generalized to Western nations with low inequality. In general, research on subjective well-being has been conducted mostly in Western nations, and it is still unsure how the results can be generalized to other cultures [15]. Furthermore, studies on ethnic differences are still rare, and firm conclusions are still impossible for the impact of race [6]. Therefore, caution should be applied when comparing the results with other cultures.

The use of more than two time points could have strengthened the results, but subjective well-being was not measured in the 1998 survey. Impairment caused by a particular disease might differ considerably between individuals, but the severity was not reported in the survey. However, multimorbidity has been shown to linearly associate with life satisfaction [31] and, more generally, be an important health indicator. Grouping according to the number of reported diseases was considered a suitable way to differentiate between participants. This grouping was used in the study due to a statistically significant effect.

The dietary habits were measured by self-report, where information about special diets was not included. However, as the observed variable for dietary habits was dichotomized, individual dietary restrictions are likely to have a minor impact. Lastly, non-response and attrition has resulted in some underrepresentation of men and individuals having a lower level of education, fewer healthy behaviors, or lower subjective well-being. The length and the sensitive nature of the survey were reasons for non-response in 1998 [32] and have presumably also resulted in increasing attrition of participants during the follow-up.

## Further study

In our study, we examined the relationship of the four principal health behaviors that have substantial impact on non-communicable diseases and subjective well-being. However, the

health behaviors do not covary consistently and it is also unclear whether the included health behaviors are the principal ones associated with subjective well-being. Additional health behaviors–such as meditation, avoiding sedentary behavior, and sleep–could also have a role and would therefore be worth studying. Health behavior as a latent variable could also be studied in more detail. Additional observed variables, a larger variety of covariates, and a more detailed scale for all health behaviors could be tested, e.g., including former smokers as a separate group of smoking status. The study of the relationship between health behavior and subjective well-being would benefit from experimental interventions where habits improving subjective well-being would be encouraged to support a change in multiple health-behaviors.

## Conclusion

Our results suggest that health behavior partly predicts subjective well-being in a longitudinal follow-up. The study also underlines the stability of health behavior. These results could serve as motivators for health behavior change in health promotion.

## Supporting information

**S1 File. Comparison of the study population and respondents to the survey.** Data showing that omitting participants due to missing information on covariates does not cause major changes in the characteristics of the study population or distort the SEM model.
(DOCX)

**S2 File. Reliability measures.**
(DOCX)

## Author Contributions

**Conceptualization:** Säde Stenlund, Heli Koivumaa-Honkanen, Sakari Suominen, Hanna Lagström, Päivi Rautava.

**Data curation:** Säde Stenlund, Lauri Sillanmäki, David Stenlund.

**Formal analysis:** Säde Stenlund, Niina Junttila, David Stenlund.

**Funding acquisition:** Säde Stenlund.

**Investigation:** Säde Stenlund, Heli Koivumaa-Honkanen, Sakari Suominen, Päivi Rautava.

**Methodology:** Säde Stenlund, Niina Junttila, Heli Koivumaa-Honkanen, Lauri Sillanmäki, David Stenlund, Sakari Suominen, Päivi Rautava.

**Project administration:** Säde Stenlund.

**Resources:** Heli Koivumaa-Honkanen, Sakari Suominen, Päivi Rautava.

**Supervision:** Heli Koivumaa-Honkanen, Päivi Rautava.

**Visualization:** Säde Stenlund, David Stenlund.

**Writing – original draft:** Säde Stenlund.

**Writing – review & editing:** Säde Stenlund, Niina Junttila, Heli Koivumaa-Honkanen, Lauri Sillanmäki, David Stenlund, Sakari Suominen, Hanna Lagström, Päivi Rautava.

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
