## [Decision Letter · Decision Letter 0]

21 Jun 2021

PONE-D-21-14933

Longitudinal stability and interrelations between health behavior and subjective well-being in a follow-up of nine years

PLOS ONE

Dear Authors,

Thank you for submitting your manuscript to PLOS ONE. After careful consideration, we feel that it has merit but does not fully meet PLOS ONE’s publication criteria as it currently stands. Therefore, we invite you to submit a revised version of the manuscript that addresses the points raised during the review process.

Please submit your revised manuscript by 17 July 2021. If you will need more time than this to complete your revisions, please reply to this message or contact the journal office at plosone@plos.org. Please include the following items when submitting your revised manuscript:

We look forward to receiving your revised manuscript.

Kind regards,

Marcel Pikhart

Academic Editor

PLOS ONE

Journal Requirements:

3. We noted in your submission details that a portion of your manuscript may have been presented or published elsewhere. 

"The data has been used in two other manuscripts exploring the relationship between health behavior and life satisfaction previously by linear regression models and sum scores of health behavior and subjective well-being. In the present study a more intricate method is applied to gain deeper understanding of the relationship."

Please clarify whether this publication was peer-reviewed and formally published. If this work was previously peer-reviewed and published, in the cover letter please provide the reason that this work does not constitute dual publication and should be included in the current manuscript.

Reviewers' comments:

Reviewer's Responses to Questions

**Comments to the Author**

1. Is the manuscript technically sound, and do the data support the conclusions?

Reviewer #1: Yes

Reviewer #2: Partly

2. Has the statistical analysis been performed appropriately and rigorously? 

Reviewer #1: Yes

Reviewer #2: Yes

3. Have the authors made all data underlying the findings in their manuscript fully available?

Reviewer #1: Yes

Reviewer #2: No

4. Is the manuscript presented in an intelligible fashion and written in standard English?

Reviewer #1: Yes

Reviewer #2: Yes

5. Review Comments to the Author

Reviewer #1: This longitudinal follow-up research is promising. There are some critical issues whose resolution will increase the strength of the manuscript and contribute to its publication, as mentioned below:

-You have cited a lot of reference based on participants from different nationalities such as US, UK, and so on. Do different races have an impact on the research results?

-The grouping of participants in this study is confusing to some extent, for example, the age grouping in not continuous, and the major diseases grouping is based on the number rather than the type of disease. Is there any reason for this grouping?

-When studying smoking status, you have only divided it into current smokers and others, while the time after quitting smoking may make a difference in health behavior. Could you divide this group into several subgroups?

-The response rate of Survey (2003) sent to persons who responded in 1998 was 80.2% while the response rate of Survey (2012) sent to persons who responded in 1998 was 57.4%. It is interesting to find this big difference, can you explain why were there so much censored data?

-This research has a large number of participants which adds credibility to results, could you provide a table describing the baseline information of participants?

PAGE 9, LINE 206: The R value is overwritten by other numbers.

Reviewer #2: Background

There is no clear what concept of subjective well-being was considered in the study. This should have significant implications for its operationalization.

The aim of study is presented too general. It is recommended to add more specific problems with hypotheses and graphic presentation of the tested model (here rather not later).

Methods

Were the populations of a random sample (n=13,050) and this finally included in the study (n=11,806) equal according to study variables?

Health behaviours measure: as for adherence to Nordic nutritional recommendations - how responses of people on gluten-free or lactose-free diets were classified? (was the diet controlled for?)

What was the reliability of health behaviour and subjective well-being indicators? Goodness of fit of the model (CFA) does not imply reliability of the scales. In the Discussion section, there are some considerations on stability of subjective well-being but it is not sufficient to understand the character and the role of this indicator. Kuder-Richardson’s and Cronbach’s alfa coefficients (respectively) should be presented.

Additionally, the following issue demands explanation: one of the items included in subjective well-being indicator differs in the scale of responses (criterion for creating psychometrically correct indicators).

It would be also valuable if the Authors explain what was the aim and the effect of inclusion of autocorrelations in the model.

Results

Table 2 – please check the title and content (should it variance or SD?)

I couldn’t find chi2 tests for models’ comparisons.

Discussion

In the Discussion section, there is excessive and not supported by data concentration on the effect of health behavior in 2003 on subjective well-being in 2012, ignoring the fact the this relationship is very week (especially taking into account the sample size). It is rather difficult to treat it as a factor encouraging to health behaviour change. It seems also that health behaviour indicator has low consistency and maybe it would be better to check the effect of individual health behaviors on subjective well-being (assuming that reliability of this scale is satisfactory).

There are single editing and typing errors (e.g. line 130).

6. PLOS authors have the option to publish the peer review history of their article (what does this mean?). If published, this will include your full peer review and any attached files.

Reviewer #1: No

Reviewer #2: No

---

## [Author Response · Author response to Decision Letter 0]

20 Aug 2021

For better layout, see the attached Response to reviewers.

Dear Marcel Pikhart, 

Thank you for the feedback on our manuscript. We found the reviewer comments helpful for improving the manuscript. Below we have addressed all the comments (numbered) point by point (in bold) and included the new text (in blue) also after the response. The pages and lines mentioned refers to the final version without any visible changes. 

JOURNAL REQUIREMENTS:

Comment 1: Please ensure that your manuscript meets PLOS ONE’s style requirements, including those for file naming. 

Response 1: The style instructions resulted in the following changes: 

a) Background section was renamed as Introduction on page 3, line 59. 

b) Supporting information captions were included as follows on page 22, line 506. 

Supporting information 

S1 File. Comparison of the study population and respondents to the survey. Data showing that omitting participants due to missing information on covariates does not cause major changes in the characteristics of the study population or distort the SEM model. 

S2 File. Reliability measures.

Comment 2: We note that you have indicated that data from this study are available upon request. In your revised cover letter, please address the following prompts:

Response 2: The following information was updated to the cover letter: 

… study data contains personal and sensitive information and due to the present legislation in Finland cannot be made publicly available inside or outside Finland. Lauri Sillanmäki (lauri.sillanmaki@utu.fi) is responsible for the data storage and can be contacted concerning data inquiries.

Comment 3. We noted in your submission details that a portion of your manuscript may have been presented or published elsewhere. "The data has been used in two other manuscripts exploring the relationship between health behavior and life satisfaction previously by linear regression models and sum scores of health behavior and subjective well-being. In the present study a more intricate method is applied to gain deeper understanding of the relationship."

Please clarify whether this publication was peer-reviewed and formally published. If this work was previously peer-reviewed and published, in the cover letter please provide the reason that this work does not constitute dual publication and should be included in the current manuscript.

Response 3: The following information was updated to the cover letter: 

Two manuscripts have been submitted using the current data. The first has been peer-reviewed and resubmitted waiting for final decision. The second manuscript was suggested to be transferred to another journal according to the peer-reviewers’ comments. Both of them use general linear regression models whereas the current manuscript uses a more complex method i.e. structural equation modeling (SEM). In SEM, each component of health behavior and subjective well-being have their distinct impact in contrast to the sum scores used in the previous two manuscripts with only linear regression analyses. SEM enables the inclusion of the cross-sectional, cross-lagged, and longitudinal relationships between health behavior and subjective well-being at both baseline and at follow-up. Therefore, SEM is a more advanced method for detailed analysis. It provides a different perspective and gives deeper knowledge compared to our earlier manuscripts. Thus, the present manuscript gives further knowledge on the studied relationship not given by the previous manuscript.

REVIEWER REPORTS:

REVIEWER 1: 

Comment 4: This longitudinal follow-up research is promising. There are some critical issues whose resolution will increase the strength of the manuscript and contribute to its publication, as mentioned below:

-You have cited a lot of reference based on participants from different nationalities such as US, UK, and so on. Do different races have an impact on the research results?

Response 4: Most studies on subjective well-being are conducted in Western nations and generalizability to other nations is still unsure. As a Western nation, Finland has a strong social security system and low inequality, which might affect the results. However, generalizability between Western nations is usually considered acceptable, although differing characteristics of the nations where the results are obtained is important to take into account. An elaboration was added in the evaluation section as follows on page 17 line 371: 

“The study was conducted in Finland, where the social security system is strong and for example gender or economic inequality low. Therefore, the results are most reliably generalized to Western nations with low inequality. In general, research on subjective well-being has been conducted mostly in Western nations, and it is still unsure how the results can be generalized to other cultures [27]. Furthermore, studies on ethnic differences are still rare, and firm conclusions are still impossible for the impact of race [6]. Therefore, caution should be applied when comparing the results with other cultures.”

Comment 5: The grouping of participants in this study is confusing to some extent, for example, the age grouping in not continuous, and the major diseases grouping is based on the number rather than the type of disease. Is there any reason for this grouping?

Response 5: The following clarification on age groups was added on page 5 line 119: 

“When the HeSSup-study commenced in 1998, the age groups were chosen to be non-continuous to capture certain life transition periods by creating clearly distinct groups that would be fairly homogeneous, i.e., ages not spreading over an entire decade.”

An evaluation of the covariate of diseases was included on page 17 line 380: 

“Impairment caused by a particular disease might differ considerably between individuals, but the severity was not reported in the survey. However, multimorbidity has been shown to linearly associate with life satisfaction [31] and, more generally, be an important health indicator. Grouping according to the number of reported diseases was considered a suitable way to differentiate between participants. This grouping was used in the study due to a statistically significant effect.”

Comment 6: When studying smoking status, you have only divided it into current smokers and others, while the time after quitting smoking may make a difference in health behavior. Could you divide this group into several subgroups?

Response 6: When preparing the first version of the manuscript, smoking was a three-category variable, but it was converted into a two-category variable to conform with the other health behavior indicators, all of which were dichotomized. Mixing dichotomized and other categorical observed variables connected to the same latent variable is possible in SEM, but not preferable. Furthermore, previous research analyzing multiple health behaviors frequently uses dichotomized health behaviors. Exploring more detailed health behavior scales could be a possible direction for future research. A short addition was included in the text on page X line X. 

“Additional observed variables, a larger variety of covariates and a more detailed scale for all health behaviors could be tested, e.g., including former smokers as a separate group of smoking status.”

Comment 7: The response rate of Survey (2003) sent to persons who responded in 1998 was 80.2% while the response rate of Survey (2012) sent to persons who responded in 1998 was 57.4%. It is interesting to find this big difference, can you explain why were there so much censored data?

Response 7: The survey was long: 29 pages containing 112 detailed and personal items. Complying to filling out such a survey requires effort and the probability of not responding increases as the number of waves increase. The following text was added to the manuscript page 17 line 389: 

“Lastly, non-response and attrition has resulted in some underrepresentation of men and individuals having a lower level of education, fewer healthy behaviors, or lower subjective well-being. The length and the sensitive nature of the survey were reasons for non-response in 1998 [32] and have presumably also resulted in increasing attrition of participants during the follow-up.” 

Comment 8: This research has a large number of participants which adds credibility to results, could you provide a table describing the baseline information of participants?

Response 8: A table is provided with baseline information about the participants. The corresponding percentages were removed from the text in the methods section. A reference to the table is provided on page 9 lines 211: 

“The baseline characteristics of the participants are presented in Table 1.”

Table 1. Baseline characteristics of the participants in 2003.

Variable Category Share of the study population

% (n)

Study population 100 (11,806)

Age 25–29 20.7 (2,449)

 35–39 20.5 (2,422)

 45–49 26.7 (3,155)

 55–59 32.0 (3,780)

Gender Male 37.1 (4,382)

 Female 62.9 (7,424) 

Education No professional education 12.2 (1,436)

 Vocational school 28.7 (3,391)

 College 39.0 (4,599)

 University level education 20.2 (2,380)

Diseases 0 18.0 (2,129)

 1 23.4 (2,759)

 2 or more 58.6 (6,918)

Comment 9: PAGE 9, LINE 206: The R value is overwritten by other numbers. 

Response 9: The line numbers had overwritten the R-values and therefore the table has now been adjusted on page 10 and line 234.

REVIEWER 2: 

Background

Comment 10: There is no clear what concept of subjective well-being was considered in the study. This should have significant implications for its operationalization.

Response 10: Thank you for reviewing the manuscript and for the comments. The concept of subjective well-being was added on page 3 line 70: 

“Subjective well-being refers to a personal evaluation and appraisal of one’s life including both a cognitive judgement (such as life satisfaction) and an emotional response on life (such as happiness) [15].”

Comment 11: The aim of study is presented too general. It is recommended to add more specific problems with hypotheses and graphic presentation of the tested model (here rather not later).

Response 11: The aim was specified and the hypothesis stated on page 4, line 91. 

“The aim of the present study was to explore the cross-sectional, longitudinal, and cross-lagged relationships of health behavior and subjective well-being by structural equation modelling, as shown in Fig 1. Our hypothesis was that health behavior predicts subsequent subjective well-being and vice versa, because bidirectionality of the relationship has been suggested earlier [14]. Based on earlier research, we also anticipated that the cross-sectional relationships between health behavior and subjective well-being would be statistically significant. Furthermore, we presumed that health behavior at baseline would be a significant predictor of health behavior at follow-up, as well as subjective well-being predicting subsequent subjective well-being.”

Additionally, the graphic representation of the tested model (Figure 1) was moved below the aims as suggested. 

Methods 

Comment 12: Were the populations of a random sample (n=13,050) and this finally included in the study (n=11,806) equal according to study variables? 

Response 12: The close similarity between the population used in the final model and the total respondents of the two waves is shown in two ways on page 5 lines 111 and page 8, line 193: 

“The selection process of the study population is outlined in Fig 2. The final study population (n = 11,806) did not differ considerably from the respondents of the two waves (n = 13,050) with regard to age and gender. For details of the comparison, see S1 File.” 

“This crude model was tested for the respondents of the two waves (n = 13,050) and for the population used in the final model (n = 11,806), which confirmed that the omission of participants having missing data on covariates did not substantially alter the results. For details, see S1 File.“

S1 File displays data on these comparisons.

Comment 13: Health behaviours measure: as for adherence to Nordic nutritional recommendations - how responses of people on gluten-free or lactose-free diets were classified? (was the diet controlled for?)

Response 13: The dietary measure did not take into account special diets. However, as the dietary index contains information on 10 food items and a special diet usually has major impact on only one, the impact of special diets is not seen as a major concern. Furthermore, in Finland lactose- and gluten-free products are readily available and therefore does not necessarily have a large impact. An additional comment on special diets was included on page 17 line 387: 

“The dietary habits were measured by self-report, where information about special diets was not included. However, as the observed variable for dietary habits was dichotomized, individual dietary restrictions are likely to have a minor impact.”

Comment 14: What was the reliability of health behaviour and subjective well-being indicators? Goodness of fit of the model (CFA) does not imply reliability of the scales. In the Discussion section, there are some considerations on stability of subjective well-being but it is not sufficient to understand the character and the role of this indicator. Kuder-Richardson’s and Cronbach’s alfa coefficients (respectively) should be presented.

Response 14: Kuder-Richardson (KR-20) and Cronbach’s alpha estimates were computed using SPSS. However, since the factors are not tau-equivalent, the validity of these coefficients is questionable, and the values are therefore included in S2 File together with some comments. Instead, the omega coefficient described in S2 File is preferable. The omega values were calculated using Mplus and are given in S2 File. This was clarified shortly in the a) Statistical analysis section on page 9 line 204 and b) Discussion section in on page 13 line 284:

a) “A discussion on reliability measures is included in S2 File. Since the assumption of tau-equivalence does not hold, McDonald’s ω is preferable over the commonly used Cronbach’s α as a measure of reliability. Values of both α and ω are given in Table A in S2 File.”

b) “This suggests that the components of subjective well-being are more consistently co-varying than those of health behavior, which is also reflected in Cronbach’s alphas being higher for subjective well-being than for health behavior. Note, however, the caveats of using Cronbach’s alpha discussed in S2 File. “ 

In addition, the stability of health behavior and subjective well-being respectively was clarified in two places i.e. a) on page 13 line 287 and b) on page 17 line 368: 

a) “However, subjective well-being was a less stable characteristic during follow-up than the latent variable of health behavior when using continuous measures for subjective well-being and dichotomized for health behavior.” 

b) Health behavior was a more stable characteristic than subjective well-being, but it was measured by dichotomized variables, where small changes remain unrecorded compared to the scales for subjective well-being.

Lastly, the focus in the current study was to explore the effect of the four principal health behaviors on subjective well-being. Therefore, these four were included in the model, even though the fit or reliability could probably be improved by a different selection of health behaviors. To underline this focus, clarifying additions were provided a) and b) in the abstract (page 2, lines 34 and 38), c) in the beginning of the discussion (on page 13, line 269). The issue was further addressed d) in the evaluation section (on page 16, line 361) and e) in future research (page 18, line 397): 

a) “In the present study, we deepen this knowledge focusing on the four principal health behaviors and using structural equation modeling with selected covariates.”

b) “Structural equation modeling was used to study the cross-sectional, cross-lagged, and longitudinal relationships between the four principal health behaviors and subjective well-being at baseline and after the nine-year follow-up adjusted for age, gender, education, and self-reported diseases.”

c) “Our study on 11,800 working-age Finns was focused on the association between four principal health behaviors and subjective well-being as latent variables in a structural equation model adjusted for age, gender, education and major diseases at baseline.”

d) “The four principal health behaviors associated to the risk of chronic diseases of public health concern [29,30] were the focus of this study. However, their factor loadings varied considerably. Different components of health behavior or individual health behaviors could yield more reliable factors and results.”

e) “However, the health behaviors do not covary consistently and it is also unclear whether the included health behaviors are the principal ones associated with subjective well-being.”

Comment 15: Additionally, the following issue demands explanation: one of the items included in subjective well-being indicator differs in the scale of responses (criterion for creating psychometrically correct indicators).

Response 15: A clarification on the item of loneliness was included on page 7, line 154. In SEM, observed variables of differing scales can be attributed to the same latent factor. 

“On the original scale, feelings of loneliness ranged from 1 to 5 but had no response alternative 2. The scale was compressed into a scale ranging from 1 to 4.”

Comment 16: It would be also valuable if the Authors explain what was the aim and the effect of inclusion of autocorrelations in the model.

Response 16: This is now clarified in the evaluation section on page 16, line 353. 

“Including autocorrelation accounted for measurement error, which is of importance especially for the continuous measure of subjective well-being, resulting in more reliable estimates of variance.”

Results

Comment 17: Table 2 – please check the title and content (should it variance or SD?)

Response 17: In Table 2 (now Table 3) the variance was reported, as stated in the title. We changed this and now the standard deviation is reported instead on page 11, line 246.

Table 3. Means (standard deviations) of observed variables in the Health and Social Support (HeSSup) study.

 2003 2012

Health behavior

Risky = 1

Beneficial = 2 Physical activity (1 or 2) 1.72 1.71 

 Dietary habits (1 or 2) 1.39 1.47 

 Alcohol consumption (1 or 2) 1.95 1.95 

 Smoking status (1 or 2) 1.81 1.86 

Subjective well-being Interest in life (1–5) 3.90 (0.93) 3.88 (0.92)

 Happiness in life (1–5) 3.97 (0.83) 3.98 (0.82)

 Ease of living (1–5) 3.47 (1.02) 3.63 (0.99)

 Not feeling lonely (1–4) 3.43 (0.91) 3.46 (0.89)

Comment 18: I couldn’t find chi2 tests for models’ comparisons.

Response 18: �2 tests comparing the different models were not performed, but the �2 values for the individual models are given in Table 4 (earlier Table 3). A reformulation on �2 was provided on page 7, line 172:

“The �2 values were not primarily used for model fit estimation, because a good fit using �2 can be hard to achieve in large samples [26], but the �2 values were still used for comparison between different models.”

Discussion

Comment 19:

In the Discussion section, there is excessive and not supported by data concentration on the effect of health behavior in 2003 on subjective well-being in 2012, ignoring the fact the this relationship is very week (especially taking into account the sample size). It is rather difficult to treat it as a factor encouraging to health behaviour change. It seems also that health behaviour indicator has low consistency and maybe it would be better to check the effect of individual health behaviors on subjective well-being (assuming that reliability of this scale is satisfactory).

Response 19: The a) abstract, b) discussion , c) implications, d) evaluation and e) conclusion, have been adjusted according to the reviewer’s views on the association between health behavior in 2003 and subjective well-being in 2012. Some elaboration on future research is also included. 

a) Abstract, page 2, line 50: “The four principal health behaviors together predict subsequent subjective well-being after an extensive follow-up. Although not particularly strong, the results could still be used for motivation for health behavior change, because of the beneficial effects of health behavior on subjective well-being.”

b) Discussion, page 13, line 271: “The results suggest that health behavior predicts subjective well-being after a nine-year follow-up at a weak but still significant level, but not vice versa.”

c) Implications, page 15, line 336: “The results could also emphasize the relevance of political actions targeting health behavior, not only to reduce the increasing costs of non-communicable diseases, but also to maintain and improve people’s subjective well-being.”

d) Evaluation, page 16, line 357: “The association between health behavior and subsequent subjective well-being was statistically significant but fairly weak. However, its strength was about a third of the path coefficient of how subjective well-being predicted its subsequent level and of similar magnitude to the results in the study by Lappan et al. [17] exploring the relationship between smoking and measures of subjective well-being.”

e) Conclusion, page 18, line 409: Our results suggest that health behavior partly predicts subjective well-being in a longitudinal follow-up. The study also underlines the stability of health behavior. These results could serve as motivators for health behavior change in health promotion. “

Comment 20: There are single editing and typing errors (e.g. line 130).

Response 20: The typing error mentioned and a few more have been corrected. 

i.e. “on loneliness”and “influence of several confounders was taken into consideration”

Additional journal requirements: 

Comment 21: Thank you for including your ethics statement on the online submission form: "The concurrent joint Ethical Committee of the University of Turku and the Turku University Central Hospital approved the Health and Social Support study. The present study was carried out according to the Declaration of Helsinki. Participants signed a written consent agreeing to a prospective follow-up including the registry data." To help ensure that the wording of your manuscript is suitable for publication, would you please also add this statement at the beginning of the Methods section of your manuscript file.

Response 21: The paragraph was added in the first part of methods section. On page 5, lines 127. 

Comment 22: In line with our goal of ensuring long-term data availability to all interested researchers, PLOS’ Data Policy states that authors cannot be the sole named individuals responsible for ensuring data access (http://journals.plos.org/plosone/s/data-availability#loc-acceptable-data-sharing-methods).

Therefore, please provide a non-author, point of contact where others can request access to your minimal data set.

Response 22: Prof. Markku Sumanen (markku.sumanen@tuni.fi), leader of the HeSSup research group, agreed to be a non-author contact person for the data, which was also communicated to the journal through the submission system.

---

## [Decision Letter · Decision Letter 1]

18 Oct 2021

Longitudinal stability and interrelations between health behavior and subjective well-being in a follow-up of nine years

PONE-D-21-14933R1

Dear Authors,

We’re pleased to inform you that your manuscript has been judged scientifically suitable for publication and will be formally accepted for publication once it meets all outstanding technical requirements.

Kind regards,

Marcel Pikhart

Academic Editor

PLOS ONE

Additional Editor Comments (optional):

Reviewers' comments:

Reviewer's Responses to Questions

**Comments to the Author**

1. If the authors have adequately addressed your comments raised in a previous round of review and you feel that this manuscript is now acceptable for publication, you may indicate that here to bypass the “Comments to the Author” section, enter your conflict of interest statement in the “Confidential to Editor” section, and submit your "Accept" recommendation.

Reviewer #1: All comments have been addressed

Reviewer #2: All comments have been addressed

2. Is the manuscript technically sound, and do the data support the conclusions?

Reviewer #1: (No Response)

Reviewer #2: Yes

3. Has the statistical analysis been performed appropriately and rigorously? 

Reviewer #1: (No Response)

Reviewer #2: Yes

4. Have the authors made all data underlying the findings in their manuscript fully available?

Reviewer #1: (No Response)

Reviewer #2: Yes

5. Is the manuscript presented in an intelligible fashion and written in standard English?

Reviewer #1: (No Response)

Reviewer #2: Yes

6. Review Comments to the Author

Reviewer #1: (No Response)

Reviewer #2: The introduced changes correctly completed the missing information and contributed to the improvement of the quality of the manuscript. I have no further questions or comments.

7. PLOS authors have the option to publish the peer review history of their article (what does this mean?). If published, this will include your full peer review and any attached files.

Reviewer #1: No

Reviewer #2: No

---

## [Editor Report · Acceptance letter]

22 Oct 2021

PONE-D-21-14933R1 

Longitudinal stability and interrelations between health behavior and subjective well-being in a follow-up of nine years 

Dear Dr. Stenlund:

I'm pleased to inform you that your manuscript has been deemed suitable for publication in PLOS ONE. Congratulations! Your manuscript is now with our production department. 

Kind regards, 

on behalf of

Dr. Marcel Pikhart 

Academic Editor

PLOS ONE